# Diagnostic Usefulness of Diffusion-Weighted MRI for Axillary Lymph Node Evaluation in Patients with Breast Cancer

**DOI:** 10.3390/diagnostics13030513

**Published:** 2023-01-31

**Authors:** Pyeonghwa Cho, Chang Suk Park, Ga Eun Park, Sung Hun Kim, Hyeon Sook Kim, Se-Jeong Oh

**Affiliations:** 1Department of Radiology, Incheon St. Mary’s Hospital, College of Medicine, The Catholic University of Korea School of Medicine, Seoul 21431, Republic of Korea; 2Department of Radiology, Seoul St. Mary’s Hospital, College of Medicine, The Catholic University of Korea School of Medicine, Seoul 06591, Republic of Korea; 3Department of General Surgery, Incheon St. Mary’s Hospital, College of Medicine, The Catholic University of Korea School of Medicine, Seoul 21431, Republic of Korea

**Keywords:** breast cancer, axilla, lymph nodes, diffusion-weighted imaging, apparent diffusion coefficient (ADC)

## Abstract

This study aimed to determine whether apparent diffusion coefficient (ADC) and morphological features on diffusion-weighted MRI (DW-MRI) can discriminate metastatic axillary lymph nodes (ALNs) from benign in patients with breast cancer. Two radiologists measured ADC, long and short diameters, long-to-short diameter ratio, and cortical thickness and assessed eccentric cortical thickening, loss of fatty hilum, irregular margin, asymmetry in shape or number, and rim sign of ALNs on DW-MRI and categorized them into benign or suspicious ALNs. Pathologic reports were used as a reference standard. Statistical analysis was performed using the Mann–Whitney U test and chi-square test. Overall sensitivity, specificity, positive predictive value (PPV), negative predictive value (NPV), and diagnostic accuracy of DW-MRI were calculated. The ADC of metastatic ALNs was 0.905 × 10^−3^ mm^2^/s, and that of benign ALNs was 0.991 × 10^−3^ mm^2^/s (*p* = 0.243). All morphologic features showed significant difference between the two groups. The sensitivity, specificity, PPV, NPV, and diagnostic accuracy of the final categorization on DW-MRI were 77.1%, 93.3%, 79.4%, 92.5%, and 86.2%, respectively. Our results suggest that morphologic evaluation of ALNs on DWI can discriminate metastatic ALNs from benign. The ADC value of metastatic ALNs was lower than that of benign nodes, but the difference was not statistically significant.

## 1. Introduction

In breast cancer patients, the extent of axillary lymph node (ALN) metastasis and tumor size are two of the most important prognostic indicators for survival [1]. The regional lymph node (LN) status influences treatment options, extent of surgery, and need for chemotherapy or radiation therapy. The 5-year survival rate of breast cancer patients with regional LN metastasis is lower than that of breast cancer patients without LN metastasis [2].

For axillary nodal staging, sentinel lymph node biopsy is the standard method, followed by axillary dissection [3,4]. However, these two methods are associated with morbidity such as arm pain, seroma, hematoma, lymphedema, tingling sensations, and numbness [5]. As a result, non-invasive nodal staging techniques with appropriate diagnostic accuracy are in high demand. The non-invasive diagnostic alternatives, including physical examination, digital mammography, ultrasonography, computed tomographic scan (CT), positron emission tomographic imaging (PET), and magnetic resonance imaging (MRI), are currently used and show variable success in predicting lymph node involvement [6]. However, physical examination of the axilla showed a high false-negative rate. While axillary US is routinely used in many centers, it shows a variable sensitivity of 45.2–86.2% and variable specificity of 40.5–86.6% [6,7,8,9,10].

MRI is regarded as the best diagnostic modality to reveal the anatomy and evaluate axillary node status. On contrast-enhanced MRI, suspicious nodal morphologic features were cortical thickening, loss of fatty hilum, heterogeneous enhancement, round shape, and low long-to-short diameter ratio (L/S ratio) [9]. Diffusion-weighted imaging (DWI) is a valuable MRI sequence for distinguishing benign and malignant lesions without contrast agents. This sequence provides useful information by quantifying the restriction experienced by water in regions associated with malignancies and demands reliable mapping of ADC or other diffusivity-related parameters [11]. This sequence has no need for contrast agents and is advantageous in cases of contrast allergy or renal insufficiency and can shorten test times, lower patient cost, and increase accessibility. However, limited studies have examined DWI sequences with 3T MRI in regard to ADC value and morphological analysis of ALNs in breast cancer patients [12,13].

This study aimed to determine whether the apparent diffusion coefficient (ADC) and morphological features on diffusion-weighted MRI (DW-MRI) can discriminate metastatic axillary lymph nodes (ALNs) from benign in patients with breast cancer.

## 2. Material and Methods

### 2.1. Patients

This retrospective study was approved by our institutional ethical committee, which waived the requirement for written informed consent from patients.

The study group included 172 patients who were diagnosed with invasive breast cancer and had undergone MR imaging of breasts including DWI between May 2020 and February 2021 at our institution. Of these patients, 32 were excluded for the following reasons: subsequent neo-adjuvant chemotherapy (*n* = 24), no available DWI data due to exclusion from the field of view (FOV) on DWI (*n* = 5), MR imaging performed after LN biopsy (*n* = 2), and no available axillary LN pathology (*n* = 1).

Finally, a total of 140 cases from 139 patients (1 patient with bilateral breast cancer) was included for the analysis in this study. The patients’ ages ranged from 26 to 89 years with a mean of 56.3 years.

### 2.2. MRI Technique

MRI examinations were performed with patients in the prone position using a Magnetom Verio 3T system (Siemens Healthcare, Erlangen, Germany) in a dedicated eight-channel phase-array coil. We obtained MR images using the following sequences in Table 1. The images were acquired before and after the injection of gadolinium DTPA (0.1 mmol/kg Gadovist; Bayer Schering Pharma, Berlin, Germany).

### 2.3. Imaging Analysis

Breast MRIs were retrospectively reviewed by two radiologists (19 years and 2 years of experience in breast MR imaging, respectively).

Suspicious morphologic features of ALNs on MRI were followed: eccentric cortical thickenings (3 mm or more), loss of fatty hilum, L/S ratio < 1.6, short diameter 10 mm or larger, irregular margin, and asymmetry either in shape or number compared with contralateral axilla [9,14,15,16]. We added “rim sign” as a suspicious feature on DW-MRI. Kang et al. defined a high peripheral signal in a breast tumor lesion in DW-MRI as a “rim sign” and suggested that the specificity of breast cancer detection is further increased when a rim sign is seen on DW-MRI [17]. Therefore, we applied this rim sign to ALNs of breast cancer patients and analyzed its presence on DWI.

Both radiologists reviewed the DWI together and chose one LN, either the most suspicious one or the most prominent one on DWI. They measured the long diameter, short diameter, L/S ratio, eccentric cortical thickening, and ADC value of ALNs on DWI. Regions of interests (ROIs) of ADC were drawn manually in the largest and most prominent ALN, avoiding the fatty hilum (Figure 1 and Figure 2). They also evaluated eccentric cortical thickening, loss of fatty hilum, irregular margin, asymmetry either in shape or number of ipsilateral ALN compared with contralateral ALN, and rim sign of LN on DWI. The final assessment of LN was decided by consensus between the two reviewers.

The radiologists analyzed each parameter only on DWI based on b1000 image or b1200. They referred to pre-contrast T1WI only to verify visualized ALNs and avoid fatty hilum or surrounding subcutaneous fat tissue when drawing the ROI for ADC. They were blinded to histopathologic findings during assessment.

### 2.4. Statistical Analysis

Statistical comparisons were performed using the Mann–Whitney U test for continuous variables and the chi-square test for categorical variables. Radiologists’ final assessment of benign or metastatic ALNs according to DWI findings was compared with pathologic results. Among the total of 140 cases, 10 did not visualize ALNs on DWI and were included in the benign group. Cases were classified into benign or metastatic groups according to surgical pathologic results. A micrometastatic LN (tumor within a lymph node ≤ 2.0 mm) or an LN with isolated tumor cells were classified into the benign group. All statistical tests were performed in both groups.

Receiver operating characteristic (ROC) curves were used to determine the optimal ADC thresholds for discriminating benign and metastatic lymph nodes. The sensitivity, specificity, positive predictive value (PPV), and negative predictive value (NPV) of the final assessment according to DWI findings were calculated for diagnostic performance.

Statistical analyses were performed using statistical software (SPSS, version 17.0, SPSS Inc., Chicago, IL, USA). A *p*-value < 0.05 was considered to indicate statistical significance.

## 3. Results

The final histopathologic examination of 140 axillae revealed 35 cases with metastatic ALNs (25%) and 105 cases with benign ALNs (75%). The subtypes of primary breast cancer are shown in Table 2.

The mean ADC of metastatic ALNs was 0.905 × 10^−3^ mm^2^/s, and that of benign ALNs was 0.999 × 10^−3^ mm^2^/s. There was a wide range between the 25th and 75th percentiles of the ADC value of benign ALNs (Figure 3). There was no significant difference of ADC values between benign and metastatic groups (*p* = 0.185). The distribution of ADC values in benign ALNs showed an unbiased distribution and a wide range. Of the 35 malignant ALNs, only 4 had an ADC value greater than 1.1 × 10^−3^ mm^2^/s.

### 3.1. Continuous Variables

In the morphologic analysis of ALNs on the DWI sequence, long diameter, short diameter, and cortical thickness showed significant statistical differences between benign and metastatic groups. The long diameter of metastatic ALNs was significantly longer than that of benign ALNs (14.8 ± 7.7 mm vs. 10.1 ± 4.1 mm, *p* < 0.001). The short diameter of metastatic ALNs was significantly longer than that of benign ALNs (9.2 ± 6.4 mm vs. 5.1 ± 1.6 mm, *p* < 0.001). The L/S ratio of metastatic ALNs was lower than that of benign ALNs (1.8 ± 0.6 vs. 2.1 ± 0.8, *p* = 0.059). The cortex of metastatic ALNs was significantly thicker than that of benign ALNs (7.6± 7.0 mm vs. 3.2 ± 1.0 mm, *p* < 0.001) (Table 3).

Regarding the accuracy of LN size and cortical thickness, the ROC curve is presented in Figure 4, and the area under the curve (AUC) is presented in Table 4. The cortical thickness showed the highest value of 0.816, and the cut-off level for metastatic ALNs was at 4.35 mm thickness of cortex, suggesting good accuracy (*p* < 0.001).

### 3.2. Categorical Variables

In the morphologic analysis of ALNs, eccentric thickening, loss of fatty hilum, irregular margin, asymmetry either in shape or number of ipsilateral ALNs compared with contralateral ALNs, and rim sign of ALNs were evaluated on the DWI sequence. Metastatic ALNs showed significant differences in all features except the L/S ratio (*p* ≤ 0.001) (Table 3).

### 3.3. Diagnostic Performance

We compared and evaluated the correlation between final assessments of benign and metastatic ALNs on DWI and pathology results. The diagnostic performance of DWI to assess benign or metastatic ALNs was as follows: 77.1% for sensitivity, 93.3% for specificity, 79.4% for PPV, 92.5% for NPV, and 86.2% for diagnostic accuracy (Table 5).

## 4. Discussion

Our results showed no significant difference in ADC values, and significant differences in morphologic features and high diagnostic accuracy between benign and metastatic ALNs.

The mean ADC value of metastatic ALNs was lower than that of benign ALNs, but the difference was not statistically significant (Figure 3).

While the ADC value plays an effective role in discriminating between breast cancer and benign breast lesions, the meaning and usefulness of DWI and ADC values of ALNs remain controversial. In previous studies, the mean ADC of breast cancer was lower than that of benign breast tissue, ranging from 0.75 × 10^3^ mm^2^/s to 1.50 × 10^3^ mm^2^/s [18,19,20,21]. The mean ADC of metastatic ALNs ranged from 0.787 × 10^−3^ mm^2^/s to 1.182 × 10^−3^ mm^2^/s in other reports [22,23,24,25]. These studies suggested cut-off values differentiating benign and metastatic ALNs ranging from 0.889 × 10^−3^ mm^2^/s to 1.351 × 10^−3^ mm^2^/s and showed variable sensitivities (75.8%~94.7%) and specificities (70.1%~93%) [22,23,24,25].

In our study, the mean ADC of metastatic ALNs was 0.905 × 10^−3^ mm^2^/s, which was similar to previous studies [22,23,24,25]. However, the mean ADC of benign ALNs (0.999 × 10^−3^ mm^2^/s) was lower than that from previous studies, which ranged from 1.043 × 10^−3^ mm^2^/s to 1.551 × 10^−3^ mm^2^/s [22,23,24,25]. In addition, there was a wide range between the 25th and 75th percentiles of the ADC values of benign ALNs. We speculate that the ADC measurement of a small benign ALN would be less accurate due to the small size, thin cortex of benign LN, or failure to avoid the fatty hilum or surrounding subcutaneous fat. We also speculate that this result may be from our grouping of micrometastatic LNs as benign. Even though these were small foci of metastatic cells less than 2 mm, they might have influenced the relatively low ADC values of benign ALNs. As a result, there is no significant difference in ADC values between benign and malignant ALNs in our study.

Morphologic characterization of breast mass is important for differentiating malignant mass from benign tissue. Dynamic contrast-enhanced MRI is useful, while DWI is limited for assessing morphologic features of breast masses because it shows low spatial resolution and artifacts. Instead, DWI yields functional measures of the tumor microenvironment such as cell density, membrane integrity, and microstructures and is represented by ADC values [26,27]. Studies about morphologic assessment of ALNs on DWI are rare. Although morphologic evaluation of DWI was usually not performed in breast mass or ALNs, we tried to measure the long and short diameters and cortical thickness of ALNs and to assess variable morphologic features, including eccentric thickening, loss of fatty hilum, irregular margin, asymmetry in either shape or number, and rim sign on DWI. Our study showed significant differences in morphologic features and high diagnostic accuracy. Based on these DWI findings without contrast-enhanced MRI, two radiologists classified ALNs into benign or metastatic groups, with high NPV (92.5%) and specificity (93.3%). We guess that analysis of morphology of ALNs on DWI—long and short diameters, cortical thickness of ALNs, eccentric thickening, loss of fatty hilum, irregular margin, asymmetry in either shape or number, and rim sign—would be less affected than that of breast mass, which needs high resolution for characterizing detailed margin and shape. Moreover, MRI at 3.0T may yield improved signal-to-noise ratios for DWI.

He et al. [23] reported higher diagnostic performance using conventional MRI with an enhanced dynamic sequence; the diagnostic performance of the MRI-based lymph node scoring system had a sensitivity of 93% and specificity of 91%. The researchers reported overall higher sensitivity in variables of long diameter, short diameter, L/S ratio, and irregular margin. Chayakulkheeree et al. [28] also reported good diagnostic performance with a sensitivity of 98.5%, specificity of 57.8%, NPV of 96.4%, and PPV of 71%, using conventional MRI with an enhanced dynamic sequence. Compared with these previous studies, our study reported a relatively lower sensitivity (77.1%) and similar or higher specificity (93.3%). Therefore, the sensitivity of DW-MRI is slightly lower and the specificity is similar to, or rather higher than, that of conventional enhanced MRI in distinguishing benign and metastatic ALNs. Although morphologic assessment of ALNs on DWI is limited due to low spatial resolution and artifacts, it is sufficient to assess ALNs when contrast-enhanced breast MRI cannot be performed.

Our study has several limitations. The first was the high b value of DWI at b = 1000 (*n* = 66) or b = 1200 (*n* = 74), which could result in signal loss or image distortion. In addition, ADC is intrinsically dependent on many individual factors such as patient age, body temperature, tissue pressure, and perfusion rate, and on magnetic features such as MR acquisition parameters, magnetic field, and location and area of the ROI [22,29,30,31,32,33]. This might compromise the consistency of the study. Thus, the radiologists in our study referred to pre-contrast T1WI only to determine whether a visualized lesion was a true LN or not. However, these processes might lead to biased results. Second, upper ALNs may not be included in the FOV. In our study, five cases were excluded because they were not shown on the DWI. Three of the five cases were shown on mammography and four were shown on ultrasonography. Thus, a correlation with other imaging modalities is required. Third, this study was performed retrospectively in a single center and involved a small number of subjects. A larger group from multiple centers should be examined in a future prospective study.

## 5. Conclusions

Dynamic contrast-enhanced MRI in breast cancer patients has been used widely for diagnosis and staging. Morphologic evaluation of features such as margin, shape, and enhancement pattern with analysis of the kinetic curve was important in discriminating malignant from benign lesions. For such analyses, DW-MRI offers not only functional information by qualifying restriction of water motion within a tumor, but also morphologic information. Our results suggest that morphologic evaluation of ALNs on DWI can discriminate metastatic ALNs from benign without contrast enhancement. The ADC value of metastatic ALNs was lower than that of benign nodes, but the difference was not significant. The DWI for ALN evaluation may be a good alternative MR sequence to contrast-enhanced MRI, especially in cases of contrast allergy or renal insufficiency.

In the future, study of breast cancer and ALN evaluation should be extended to combinations of basic imaging modalities with new technology such as radiomics, which is an emerging technique that converts medical images into high-dimensional, mineable, and quantitative features via automatic extraction of data-characterization algorithms [34,35]. Radiomics in breast imaging has been applied to most existing modalities, including MRI, mammography, ultrasound, and digital breast tomosynthesis [36]. Radiomic research in breast imaging has shown promising results from diagnosis to prognosis [37,38,39,40,41]. Texture analysis using radiomics extracted from variable MR sequences has been used not only to differentiate between benign and malignant lesions and to distinguish between breast cancer subtypes, but also to predict treatment responses after NAC and disease-free survival. Militello et al. [39] reported that their predictive model based on 3D radiomic features extracted from dynamic contrast-enhanced MRI sequences using only the strongest enhancement phase showed good accuracy and specificity in the differentiation of malignant from benign breast lesions. Dong Y et al. [40] reported that utilization of breast cancer-specific textural features extracted from T2-weighted fat-suppression and DW-MRI improved the performance of radiomics in predicting sentinel LN metastasis without biopsy, pre-operatively. In addition, Park et al. [41] reported that the radiomics signature would be an independent biomarker for the estimation of disease-free survival in patients with invasive breast cancer.

As a future direction, we are planning to apply radiomics to non-contrast-enhanced and functional MRI such as DWI. This integration could potentially be useful for precision medicine and clinical management in patients with contrast allergy or renal insufficiency, and avoid deposition of gadolinium in the brain.

## Figures and Tables

**Figure 1 diagnostics-13-00513-f001:**
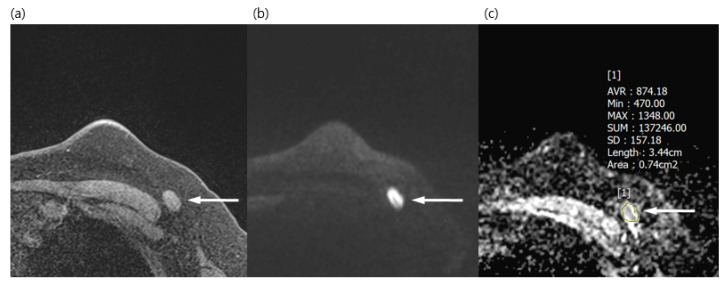
A 50-year-old woman with a surgically verified metastatic lymph node in left axilla level I. (**a**) Axial pre-contrast T1WI of the left axilla shows a 1.5 cm lymph node with loss of fatty hilum (arrow). (**b**) Axial DWI (b values, 1000 mm^2^/s) shows a lymph node (arrow) as an area of high signal intensity. (**c**) Axial ADC map (b values, 1000mm^2^/s) shows the same lesion with restricted diffusion (arrow). The region of interest was manually drawn on the ADC map, guided by pre-contrast T1WI to avoid the fatty hilum. The mean ADC of the lesion was 0.874 × 10^−3^, which indicated a metastatic lymph node.

**Figure 2 diagnostics-13-00513-f002:**
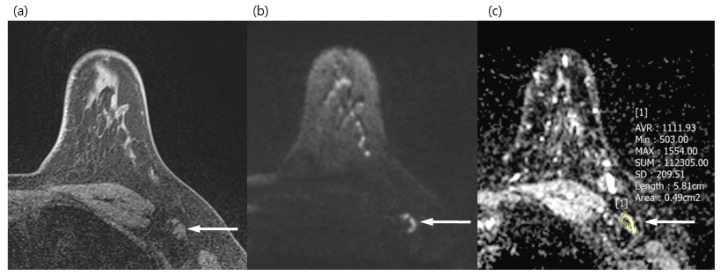
A 34-year-old woman with a surgically verified benign lymph node in left axilla level I. (**a**) Axial pre-contrast T1WI of left axilla shows a 1.2 cm lymph node with eccentric cortical thickening (arrow). (**b**) Axial DWI (b values, 1000 mm^2^/s) shows a lymph node (arrow) as a mixed area of high and low signal intensity. (**c**) Axial ADC map (b values, 1000 mm^2^/s) shows the same lesion with unrestricted diffusion (arrow). The region of interest was manually drawn on the ADC map, avoiding the fatty hilum and guided by a pre-contrast T1WI. The mean ADC of the lesion was 1.111 × 10^−3^, which indicated a benign lymph node.

**Figure 3 diagnostics-13-00513-f003:**
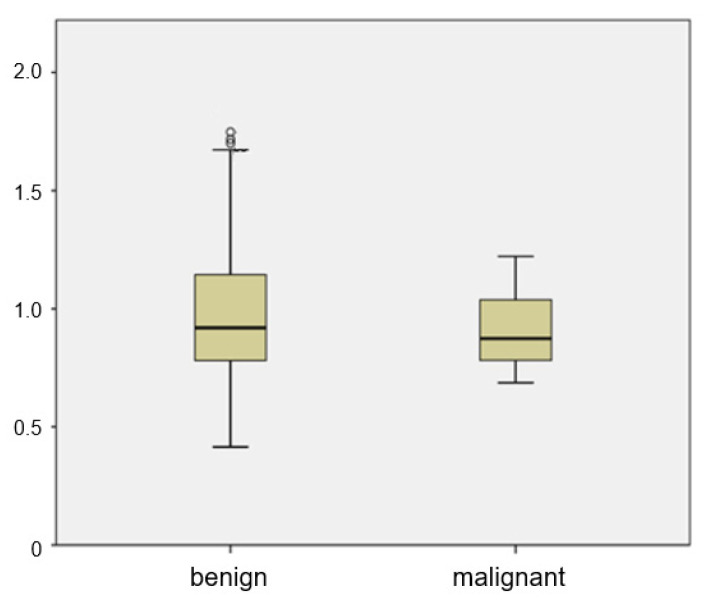
Box plot of mean ADCs of benign and metastatic ALNs. The top and bottom of each box represent the 25th and 75th percentiles of the ADC, respectively. The horizontal line inside each box represents mean value.

**Figure 4 diagnostics-13-00513-f004:**
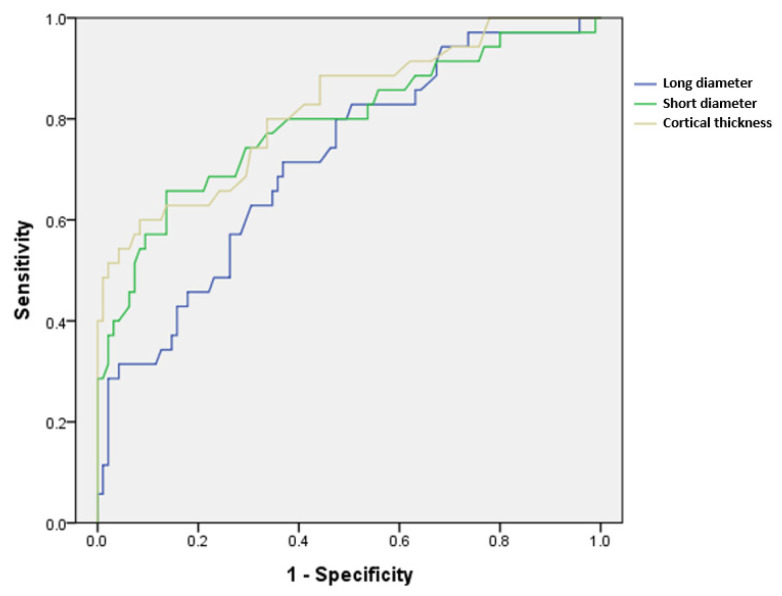
ROC curves of the continuous variables long diameter, short diameter, and cortical thickness, which showed significant differences between metastatic and benign axillary lymph nodes.

**Table 1 diagnostics-13-00513-t001:** MR protocol.

	T2WI	DWI	Pre- and Post-Contrast T1WI 3D VIBE
TR (msec)	4530	5200	2.7
TE (msec)	90	53	0.8
FOV (mm)		340 × 205	320 × 320
Matrix size	576 × 403	192 × 116	256 × 192
Slice thickness (mm)	6	4	1.2
Flip angle (°)	80		2, 6, 9, 12, 15
Acquisition time (s)	148	151	10, 70, 130, 190, 250, 310
b-value		0, 1000 or 1200	

T2WI: T2-weighted turbo spin echo sequence, DWI: diffusion-weighted sequence with readout segment echo planar imaging, T1WI 3D VIBE: T1-weighted three-dimensional volumetric interpolated breath-hold examinations, TR: repetition time, RE: echo time, FOV: field of view.

**Table 2 diagnostics-13-00513-t002:** Subtypes of primary breast cancer.

Subtypes of Primary Breast Cancer	Metastatic ALN (*n* = 35)	Benign ALN (*n* = 95)
Invasive carcinoma, no special type	29	90
Invasive lobular carcinoma	3	2
Mixed invasive carcinoma	3	2
Mucinous carcinoma	0	4
Tubular carcinoma	0	2
Metaplastic carcinoma	0	2
Encapsulated papillary carcinoma	0	2
Solid papillary carcinoma	0	1

ALN, axillary lymph node; ADC values of benign and metastatic ALNs.

**Table 3 diagnostics-13-00513-t003:** DW-MRI characteristics of axillary lymph nodes.

Variables	Benign	Malignancy	*p*
* Long diameter (mm)	10.1 ± 4.1	14.8 ± 7.7	<0.001
* Short diameter (mm)	5.1 ± 1.6	9.2 ± 6.4	<0.001
* L/S ratio	2.1 ± 0.8	1.8 ± 0.6	0.059
* Cortical thickness (mm)	3.2 ± 1.0	7.6 ± 7.0	<0.001
* ADC value (×10^−3^ mm^2^/s)	0.999 ± 0.283	0.905 ± 0.163	0.185
Eccentricity			0.001
No	68 (71.6)	14 (40)	
Yes	27 (28.4)	21 (60)	
Loss of fatty hilum			<0.001
No	94 (98.9)	20 (57.1)	
Yes	1 (1.1)	15 (42.9)	
Irregular margin			<0.001
No	93 (97.9)	25 (71.4)	
Yes	2 (2.1)	10 (28.6)	
Rim sign			<0.001
No	93 (97.9)	24 (68.6)	
Yes	2 (2.1)	11 (31.4)	
Asymmetry in either shape or number			<0.001
No	82 (86.3)	8 (22.9)	
Yes	13 (13.7)	27 (77.1)	

*: Data are mean ± standard deviation or number of patients (%). L/S: long-to-short diameter ratio of axillary lymph node, ADC: apparent diffusion coefficient.

**Table 4 diagnostics-13-00513-t004:** Area under the curve of the ROC curve in Figure 4.

Variables	AUC (95% CI)	Cut off Value	*p*
Long diameter	0.717 (0.631–0.792)	10.65 (mm)	<0.001
Short diameter	0.786 (0.706–0.853)	6.45 (mm)	<0.001
Cortical thickness	0.816 (0.739–0.879)	4.35 (mm)	<0.001
ADC value	0.577 (0.487–0.663)	1.193 (×10^−3^ mm^2^/s)	0.147
L/S ratio	0.608 (0.519–0.692)	2.15	0.045

ROC: receiver operating characteristic, AUC: area under the curve, ADC: apparent diffusion coefficient, L/S ratio: long-to-short diameter ratio of axillary lymph node.

**Table 5 diagnostics-13-00513-t005:** Sensitivity, specificity, PPV, NPV, diagnostic accuracy, and *p*-value of the following variables.

Variables	Sensitivity (%)	Specificity (%)	PPV (%)	NPV (%)	Diagnostic Accuracy (%)	*p*
Long diameter	71.4	62.1	85.5	41.0	64.6	<0.001
Short diameter	65.7	84.2	60.5	87.0	79.2	<0.001
L/S ratio	82.9	39.0	33.3	86.1	50.8	
Cortical thickness	60.0	89.5	67.7	85.9	81.5	<0.001
Eccentricity	60.0	71.6	43.8	82.9	68.5	<0.001
Loss of fatty hilum	42.9	99.0	93.8	82.5	83.9	<0.001
Irregular margin	28.6	99.0	90.9	79.0	80.0	<0.001
Asymmetry in either shape or number	80.0	76.8	56.0	91.3	83.9	0.012
Rim sign	31.4	97.9	84.6	79.5	80.0	<0.001
Final assessment	77.1	93.3	79.4	92.5	86.2	

The cut-off value of the continuous variable (long diameter, short diameter, L/S ratio, cortical thickness) was obtained through the ROC curve. The sensitivity, specificity, PPV, NPV, and diagnostic accuracy were calculated by changing the continuous variable to a categorical variable based on the cut-off value. *p* < 0.05 was considered to indicate statistical significance. L/S ratio: long-to-short diameter ratio of axillary lymph node, PPV: positive predictive value, NPV: negative predictive value.

## Data Availability

Due to privacy and institutional restrictions, the datasets analyzed during the current study are not publicly available but are available from the respective authors upon reasonable request.

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
