# Peer review of "Diagnostic Usefulness of Diffusion-Weighted MRI for Axillary Lymph Node Evaluation in Patients with Breast Cancer"

_diagnostics, 2023, doi:10.3390/diagnostics13030513_

Round 1

Reviewer 1 Report

Comment 1 (abstract)
The sentences "There was no significant differ-ence in mean ADC between metastatic and benign ALNs." and "The ADC value of metastatic ALN was lower than that of benign nodes, but the difference was not statistically significant." repeat the same concept. Please, rework them.

Comment 2 (abstract)
Should be useful to report the most representative results (e.g. sensitivity, specificity, PPV, NPV, accuracy).

Comment 3 (section 1. Introduction)
The acronym "LN" is used without it ever being defined.

Comment 4 (section 2.2. MRI technique)
To enhance readability, parameters of the MRI protocol could be reported in a table.

Comment 5 (2.4. Statistical analysis)
Please specify which analyses were done with SPSS and which with Medicalc.

comment 6 (section 3.2. Continuous variables)
In figure 4, add the extended label instead of the only acronym. In particular:
 - "LD" -> "long diameter (LD)"
 - "SD" -> "short diameter (SD)"
 - "THK" -> "cortical thickness (THK)"

Comment 7
The conclusion section is not mandatory for MDPI journals and for this reason is not present. In order to have a more complete work, I suggest adding a 'Conclusion' section summarizing the results obtained and providing possible future developments of the present work. To this aim, a possible future development of this work shoul be related to radiomics.
In fact, radiomic predictive models made possible to implement useful tools supporting decision-making of clinicians at different steps of the helthcare pipeline - e.g. lymph node metastasis predicition (Dong et al. 2018), malignancy prediction (Militello et al. 2022), or cancer recurrence prediction (Park et al. 2018) - with very interesting findings compared to conventional clinical parameters. It would be interesting to know what authors think about the possibility to implement approaches integrating clinical (such as those used in the present work) and radiomic features, in order to enhance prediction performance.
 - Dong, Y., Feng, Q., Yang, W., Lu, Z., Deng, C., Zhang, L., ... & Zhang, S. (2018). Preoperative prediction of sentinel lymph node metastasis in breast cancer based on radiomics of T2-weighted fat-suppression and diffusion-weighted MRI. European radiology, 28(2), 582-591, https://doi.org/10.1007/s00330-017-5005-7
- Militello, C., Rundo, L., Dimarco, M., Orlando, A., Woitek, R., D'Angelo, I., ... & Bartolotta, T. V. (2022). 3D DCE-MRI radiomic analysis for malignant lesion prediction in breast cancer patients. Academic Radiology, 29(6), 830-840, https://doi.org/10.1016/j.acra.2021.08.024
 - Park, H., Lim, Y., Ko, E. S., Cho, H. H., Lee, J. E., Han, B. K., ... & Park, K. W. (2018). Radiomics Signature on Magnetic Resonance Imaging: Association with Disease-Free Survival in Patients with Invasive Breast CancerRadiomics Signature on MRI for DFS in Invasive Breast Cancer. Clinical Cancer Research, 24(19), 4705-4714, https://doi.org/10.1158/1078-0432.CCR-17-3783

Author Response

Response to Reviewer 1 Comments

Point 1: (abstract)
The sentences "There was no significant differ-ence in mean ADC between metastatic and benign ALNs." and "The ADC value of metastatic ALN was lower than that of benign nodes, but the difference was not statistically significant." repeat the same concept. Please, rework them.

Point 2: (abstract)
Should be useful to report the most representative results (e.g. sensitivity, specificity, PPV, NPV, accuracy).

Response 1 & 2: Thank you for your comments. I agree that. I changed the results and conclusion in abstract. I presented the most representative results of mean ADCs of metastatic and benign axillary lymph nodes, sensitivity, specificity, PPV, NPV and diagnostic accuracy and concluded my suggestive opinion in this work.

“ADC of metastatic ALN was 0.905 x 10−3mm2/s and that of benign ALN was 0.991 x 10−3mm2/s (p = 0.243). All morphologic features showed significant difference between the two groups. The sensitivity, specificity, PPV, NPV, and diagnostic accuracy of final categorization on DW-MRI were 77.1%, 93.3%, 79.4%, 92.5%, and 86.2%, respectively. Our results suggest that morphologic evaluation of ALN on DWI can discriminate metastatic ALNs from benign. The ADC value of metastatic ALN was lower than that of benign nodes, but the difference was not statistically significant.”

Point 3: (section 1. Introduction)
The acronym "LN" is used without it ever being defined.

Response 3: Thank you for your comment. I corrected this ;”The regional lymph node (LN) status…”

Point 4: (section 2.2. MRI technique)
To enhance readability, parameters of the MRI protocol could be reported in a table.

Response 4: For easy reading my article, I added the Table about MRI protocol.

Point 5: 5 (2.4. Statistical analysis)
Please specify which analyses were done with SPSS and which with Medicalc.

Response 5: All statistics in the manuscript were performed with SPSS. The sentence has been amended as follows. In the rough draft, there were results about comparison of ROC curves performed using ‘Medcalc’. However, the content was deleted in the final draft. Thanks for the comment.

“Statistical analyses were performed using statistical software (SPSS, version 17.0, SPSS Inc., Chicago, IL, USA).”

Point 6: (section 3.2. Continuous variables)
In figure 4, add the extended label instead of the only acronym. In particular:
 - "LD" -> "long diameter (LD)"
 - "SD" -> "short diameter (SD)"
 - "THK" -> "cortical thickness (THK)"

Response 6: I changed acronyms to the extended lable in Figure 4. Thank you for the comment.

Point 7: The conclusion section is not mandatory for MDPI journals and for this reason is not present. In order to have a more complete work, I suggest adding a 'Conclusion' section summarizing the results obtained and providing possible future developments of the present work. To this aim, a possible future development of this work shoul be related to radiomics.
In fact, radiomic predictive models made possible to implement useful tools supporting decision-making of clinicians at different steps of the helthcare pipeline - e.g. lymph node metastasis predicition (Dong et al. 2018), malignancy prediction (Militello et al. 2022), or cancer recurrence prediction (Park et al. 2018) - with very interesting findings compared to conventional clinical parameters. It would be interesting to know what authors think about the possibility to implement approaches integrating clinical (such as those used in the present work) and radiomic features, in order to enhance prediction performance.
 - Dong, Y., Feng, Q., Yang, W., Lu, Z., Deng, C., Zhang, L., ... & Zhang, S. (2018). Preoperative prediction of sentinel lymph node metastasis in breast cancer based on radiomics of T2-weighted fat-suppression and diffusion-weighted MRI. European radiology, 28(2), 582-591, https://doi.org/10.1007/s00330-017-5005-7
- Militello, C., Rundo, L., Dimarco, M., Orlando, A., Woitek, R., D'Angelo, I., ... & Bartolotta, T. V. (2022). 3D DCE-MRI radiomic analysis for malignant lesion prediction in breast cancer patients. Academic Radiology, 29(6), 830-840, https://doi.org/10.1016/j.acra.2021.08.024
 - Park, H., Lim, Y., Ko, E. S., Cho, H. H., Lee, J. E., Han, B. K., ... & Park, K. W. (2018). Radiomics Signature on Magnetic Resonance Imaging: Association with Disease-Free Survival in Patients with Invasive Breast CancerRadiomics Signature on MRI for DFS in Invasive Breast Cancer. Clinical Cancer Research, 24(19), 4705-4714, https://doi.org/10.1158/1078-0432.CCR-17-3783

Response 7: Thank you for your kind comments. You suggested the new direction for next my study. Radiomics is an emerging technology these days. Although I added this point in conclusion shortly, it would be a good guidance for my next work.

“In future, the study for ALN evaluation in breast cancer patients would be extended to combination of non-contrast enhanced MRI with radiomics, an emerging technology (34)."

Reviewer 2 Report

The authors assessed the diagnostic performance of functional MR sequences (apparent diffusion coefficient -ADC and morphological features on diffusion-weighted MRI) in the setting of metastatic axillary lymph nodes in breast cancer patients. The topic is of interest, as non-invasive diagnostic procedures for axillary staging may have a relevant clinical impact in the era of treatment-de-escalation and personalized medicine. The manuscript is clear and well-written, methodology is robust and study design straightforward.

My comment is: two radiologists, with different expertise in the diagnostic assessment of breast cancer, were involved in the present study. I would ask the authors to provide a measure of the level of concordance of the 2 operators with respect to the different parameters that were assessed. I would suggest using the Cohen’s kappa coefficient to measure inter-rater reliability for qualitative items.

Author Response

Response to Reviewer 2 Comment

Point 1: two radiologists, with different expertise in the diagnostic assessment of breast cancer, were involved in the present study. I would ask the authors to provide a measure of the level of concordance of the 2 operators with respect to the different parameters that were assessed. I would suggest using the Cohen’s kappa coefficient to measure inter-rater reliability for qualitative items.

Response 1: Thank you for your ask. Because two radiologists reviewed DWI together and chose one LN by consensus, we could not measure interobserver variability. For uniform result, we thought we should measure and assess the same axillary lymph node.

Round 2

Reviewer 1 Report

Not all comments raised in the previous version of the manuscript have been resolved. Comment 7 has not been resolved.

Author Response

Response to Reviewer 1 Comments

Point 1: (abstract)
The sentences "There was no significant differ-ence in mean ADC between metastatic and benign ALNs." and "The ADC value of metastatic ALN was lower than that of benign nodes, but the difference was not statistically significant." repeat the same concept. Please, rework them.

Point 2: (abstract)
Should be useful to report the most representative results (e.g. sensitivity, specificity, PPV, NPV, accuracy).

Response 1 & 2: Thank you for your comments. I agree that. I changed the results and conclusion in abstract. I presented the most representative results of mean ADCs of metastatic and benign axillary lymph nodes, sensitivity, specificity, PPV, NPV and diagnostic accuracy and concluded my suggestive opinion in this work.

“ADC of metastatic ALN was 0.905 x 10−3mm2/s and that of benign ALN was 0.991 x 10−3mm2/s (p = 0.243). All morphologic features showed significant difference between the two groups. The sensitivity, specificity, PPV, NPV, and diagnostic accuracy of final categorization on DW-MRI were 77.1%, 93.3%, 79.4%, 92.5%, and 86.2%, respectively. Our results suggest that morphologic evaluation of ALN on DWI can discriminate metastatic ALNs from benign. The ADC value of metastatic ALN was lower than that of benign nodes, but the difference was not statistically significant.”

Point 3: (section 1. Introduction)
The acronym "LN" is used without it ever being defined.

Response 3: Thank you for your comment. I corrected this ;”The regional lymph node (LN) status…”

Point 4: (section 2.2. MRI technique)
To enhance readability, parameters of the MRI protocol could be reported in a table.

Response 4: For easy reading my article, I added the Table about MRI protocol.

Point 5: 5 (2.4. Statistical analysis)
Please specify which analyses were done with SPSS and which with Medicalc.

Response 5: All statistics in the manuscript were performed with SPSS. The sentence has been amended as follows. In the rough draft, there were results about comparison of ROC curves performed using ‘Medcalc’. However, the content was deleted in the final draft. Thanks for the comment.

“Statistical analyses were performed using statistical software (SPSS, version 17.0, SPSS Inc., Chicago, IL, USA).”

Point 6: (section 3.2. Continuous variables)
In figure 4, add the extended label instead of the only acronym. In particular:
 - "LD" -> "long diameter (LD)"
 - "SD" -> "short diameter (SD)"
 - "THK" -> "cortical thickness (THK)"

Response 6: I changed acronyms to the extended lable in Figure 4. Thank you for the comment.

Point 7: The conclusion section is not mandatory for MDPI journals and for this reason is not present. In order to have a more complete work, I suggest adding a 'Conclusion' section summarizing the results obtained and providing possible future developments of the present work. To this aim, a possible future development of this work shoul be related to radiomics.
In fact, radiomic predictive models made possible to implement useful tools supporting decision-making of clinicians at different steps of the helthcare pipeline - e.g. lymph node metastasis predicition (Dong et al. 2018), malignancy prediction (Militello et al. 2022), or cancer recurrence prediction (Park et al. 2018) - with very interesting findings compared to conventional clinical parameters. It would be interesting to know what authors think about the possibility to implement approaches integrating clinical (such as those used in the present work) and radiomic features, in order to enhance prediction performance.
 - Dong, Y., Feng, Q., Yang, W., Lu, Z., Deng, C., Zhang, L., ... & Zhang, S. (2018). Preoperative prediction of sentinel lymph node metastasis in breast cancer based on radiomics of T2-weighted fat-suppression and diffusion-weighted MRI. European radiology, 28(2), 582-591, https://doi.org/10.1007/s00330-017-5005-7
- Militello, C., Rundo, L., Dimarco, M., Orlando, A., Woitek, R., D'Angelo, I., ... & Bartolotta, T. V. (2022). 3D DCE-MRI radiomic analysis for malignant lesion prediction in breast cancer patients. Academic Radiology, 29(6), 830-840, https://doi.org/10.1016/j.acra.2021.08.024
 - Park, H., Lim, Y., Ko, E. S., Cho, H. H., Lee, J. E., Han, B. K., ... & Park, K. W. (2018). Radiomics Signature on Magnetic Resonance Imaging: Association with Disease-Free Survival in Patients with Invasive Breast CancerRadiomics Signature on MRI for DFS in Invasive Breast Cancer. Clinical Cancer Research, 24(19), 4705-4714, https://doi.org/10.1158/1078-0432.CCR-17-3783

Response 7: Thank you for your kind comments. I added conclusion section. I summarized the result of my study and suggested to plan next study, application of new technique, radiomics to non-contrast enhanced MRI. Because my present study was not about radiomics, I describe this briefly.

Conclusion

Dynamic contrast-enhanced MRI in breast cancer patients has been used widely for diagnosis and staging. Morphologic evaluation of features such as margin, shape, and enhancement pattern with analysis of the kinetic curve was important in discriminating malignant from benign lesions. For such analysis, DW-MRI offers not only functional information by qualifying restriction of water motion within a tumor, but also morphologic information. Our results suggest that morphologic evaluation of ALN on DWI can discriminate metastatic ALNs from benign without contrast enhancement. The ADC value of metastatic ALN was lower than that of benign nodes, but the difference was not significant. DWI for ALN evaluation may be a good alternative MR sequence to contrast-enhanced MRI, especially in cases of contrast allergy or renal insufficiency.

In the future, study of breast cancer and ALN evaluation should be extended to combinations of basic imaging modalities with new technology like radiomics, which is an emerging technique that converts medical images into high-dimensional, mineable, and quantitative features via automatic extraction of data-characterization algorithms 〔34,35〕. Radiomics in breast imaging has been applied to most existing modalities including MRI, mammography, ultrasound, and digital breast tomosynthesis 〔36〕. Radiomic research in breast imaging has shown promising results from diagnosis to prognosis 〔37,38〕. As a future direction, we are planning to apply radiomics to non-contrast-enhanced and functional MRI like DWI. This integration could potentially be useful for precision medicine and clinical management.

Reviewer 2 Report

I do not have any further comment

Author Response

No comment.

No response

Round 3

Reviewer 1 Report

Well done to the authors for their good work in addressing the comments received and revising the manuscript.

Previous comment 7 was only partially addressed. In order to provide a more complete view of the potential of radiomics in clinical practice - from diagnosis to prognosis - I suggest adding in Section 5. 'Conclusion' the interesting literature works suggested below:
 - Dong, Y., Feng, Q., Yang, W., Lu, Z., Deng, C., Zhang, L., ... & Zhang, S. (2018). Preoperative prediction of sentinel lymph node metastasis in breast cancer based on radiomics of T2-weighted fat-suppression and diffusion-weighted MRI. European radiology, 28(2), 582-591, https://doi.org/10.1007/s00330-017-5005-7
- Militello, C., Rundo, L., Dimarco, M., Orlando, A., Woitek, R., D'Angelo, I., ... & Bartolotta, T. V. (2022). 3D DCE-MRI radiomic analysis for malignant lesion prediction in breast cancer patients. Academic Radiology, 29(6), 830-840, https://doi.org/10.1016/j.acra.2021.08.024
 - Park, H., Lim, Y., Ko, E. S., Cho, H. H., Lee, J. E., Han, B. K., ... & Park, K. W. (2018). Radiomics Signature on Magnetic Resonance Imaging: Association with Disease-Free Survival in Patients with Invasive Breast CancerRadiomics Signature on MRI for DFS in Invasive Breast Cancer. Clinical Cancer Research, 24(19), 4705-4714, https://doi.org/10.1158/1078-0432.CCR-17-3783

Author Response

Point 1: (abstract)

Well done to the authors for their good work in addressing the comments received and revising the manuscript.

Previous comment 7 was only partially addressed. In order to provide a more complete view of the potential of radiomics in clinical practice - from diagnosis to prognosis - I suggest adding in Section 5. 'Conclusion' the interesting literature works suggested below:
 - Dong, Y., Feng, Q., Yang, W., Lu, Z., Deng, C., Zhang, L., ... & Zhang, S. (2018). Preoperative prediction of sentinel lymph node metastasis in breast cancer based on radiomics of T2-weighted fat-suppression and diffusion-weighted MRI. European radiology, 28(2), 582-591, https://doi.org/10.1007/s00330-017-5005-7
- Militello, C., Rundo, L., Dimarco, M., Orlando, A., Woitek, R., D'Angelo, I., ... & Bartolotta, T. V. (2022). 3D DCE-MRI radiomic analysis for malignant lesion prediction in breast cancer patients. Academic Radiology, 29(6), 830-840, https://doi.org/10.1016/j.acra.2021.08.024
 - Park, H., Lim, Y., Ko, E. S., Cho, H. H., Lee, J. E., Han, B. K., ... & Park, K. W. (2018). Radiomics Signature on Magnetic Resonance Imaging: Association with Disease-Free Survival in Patients with Invasive Breast CancerRadiomics Signature on MRI for DFS in Invasive Breast Cancer. Clinical Cancer Research, 24(19), 4705-4714, https://doi.org/10.1158/1078-0432.CCR-17-3783

Response 1: Thank you for your kind comments. I added the content that you suggest  three interesting articles in the conclusion section. They are about variable subjects of breast cancer imaing for personalized medicine. I summarized the result of my study and suggested to plan next study, application of new technique, radiomics to non-contrast enhanced MRI.

Thank you again for guide and give directions for my next study.

Conclusion

Dynamic contrast-enhanced MRI in breast cancer patients has been used widely for diagnosis and staging. Morphologic evaluation of features such as margin, shape, and enhancement pattern with analysis of the kinetic curve was important in discriminating malignant from benign lesions. For such analysis, DW-MRI offers not only functional information by qualifying restriction of water motion within a tumor, but also morphologic information. Our results suggest that morphologic evaluation of ALN on DWI can discriminate metastatic ALNs from benign without contrast enhancement. The ADC value of metastatic ALN was lower than that of benign nodes, but the difference was not significant. DWI for ALN evaluation may be a good alternative MR sequence to contrast-enhanced MRI, especially in cases of contrast allergy or renal insufficiency.

In the future, study of breast cancer and ALN evaluation should be extended to combinations of basic imaging modalities with new technology like radiomics, which is an emerging technique that converts medical images into high-dimensional, mineable, and quantitative features via automatic extraction of data-characterization algorithms 〔34,35〕. Radiomics in breast imaging has been applied to most existing modalities including MRI, mammography, ultrasound, and digital breast tomosynthesis 〔36〕. Radiomic research in breast imaging has shown promising results from diagnosis to prognosis 〔37-41〕. Texture analysis using radiomics extracted from variable MR sequences has been used not only to differentiate between benign and malignant lesion, distinguish between breast cancer subtypes but also to predict treatment response after NAC and disease free survivial. Militello et al. (39) reported that their predictive model based on 3D radiomic features extracted from dynamic contrast enhanced MRI sequences using only the strongest enhancement phase showed good accuracy and specificity in the differentiation of malignant from benign breast lesions. Dong Y et al. (40) reported that utilization of breast cancer-specific textural features extracted from T2-weighted fat-suppression and DW-MRI improved the performance of radiomics in predicting sentinel LN metastasis without biopsy, preoperatively. In addition, Park et al. (41) reported that the radiomics signature would be an independent biomarker for the estimation of disease-free survival in patients with invasive breast cancer.

As a future direction, we are planning to apply radiomics to non-contrast-enhanced and functional MRI like DWI. This integration could potentially be useful for precision medicine and clinical management in patients of contrast allergy or renal insufficiency and avoiding deposition of gadolinium in the brain.